# Neural Program Repair by Jointly Learning to Localize and Repair

**Marko Vasic**[1,2], **Aditya Kanade**[1,3], **Petros Maniatis**[1], **David Bieber**[1], **Rishabh Singh**[1]
[1]Google Brain, USA    [2]University of Texas at Austin, USA    [3]IISc Bangalore, India
vasic@utexas.edu  {akanade,maniatis,dbieber,rising}@google.com

## Abstract

Due to its potential to improve programmer productivity and software quality, automated program repair has been an active topic of research. Newer techniques harness neural networks to learn directly from examples of buggy programs and their fixes. In this work, we consider a recently identified class of bugs called variable-misuse bugs. The state-of-the-art solution for variable misuse enumerates potential fixes for all possible bug locations in a program, before selecting the best prediction. We show that it is beneficial to train a model that jointly and directly localizes and repairs variable-misuse bugs. We present multi-headed pointer networks for this purpose, with one head each for localization and repair. The experimental results show that the joint model significantly outperforms an enumerative solution that uses a pointer based model for repair alone.

## 1 Introduction

Advances in machine learning and the availability of large corpora of source code have led to growing interest in the development of neural representations of programs for performing program analyses. In particular, different representations based on token sequences (Gupta et al., 2017; Bhatia et al., 2018), program parse trees (Piech et al., 2015; Mou et al., 2016), program traces (Reed & de Freitas, 2015; Cai et al., 2017; Wang et al., 2018), and graphs (Allamanis et al., 2018) have been proposed for a variety of tasks including repair (Devlin et al., 2017b; Allamanis et al., 2018), optimization (Bunel et al., 2017), and synthesis (Parisotto et al., 2017; Devlin et al., 2017a).

In recent work, Allamanis et al. (2018) proposed the problem of *variable misuse* (VarMisuse): given a program, find program locations where variables are used, and predict the correct variables that should be in those locations. A VarMisuse bug exists when the correct variable differs from the current one at a location. Allamanis et al. (2018) show that variable misuses occur in practice, e.g., when a programmer copies some code into a new context, but forgets to rename a variable from the older context, or when two variable names within the same scope are easily confused. Figure 1a shows an example derived from a real bug. The programmer copied line 5 to line 6, but forgot to rename object_name to subject_name. Figure 1b shows the correct version.

Allamanis et al. (2018) proposed an *enumerative* solution to the VarMisuse problem. They train a model based on graph neural networks that learns to predict a correct variable (among all type-

```
1 def validate_sources(sources):
2   object_name = get_content(sources, 'obj')
3   subject_name = get_content(sources, 'subj')
4   result = Result()
5   result.objects.append(object_name)
6   result.subjects.append(object_name)
7   return result
```

```
1 def validate_sources(sources):
2   object_name = get_content(sources, 'obj')
3   subject_name = get_content(sources, 'subj')
4   result = Result()
5   result.objects.append(object_name)
6   result.subjects.append(subject_name)
7   return result
```

(a) An example of VarMisuse shown in red text. At test time, one prediction task is generated for each of the variable-use locations (Blue boxes).

(b) The corrected version of Figure 1a. If used at train time, one example would be generated for each of the variable-use locations (Blue boxes).

Figure 1: Enumerative solution to the VarMisuse problem.

correct variables available in the scope) for each *slot* in a program. A slot is a placeholder in a program where a variable is used. The model is trained on a synthetic dataset containing a training example for each slot in the programs from a corpus of correct source files, and teaching the model to predict the correct, existing variable for each slot. At inference time, a program of unknown correctness is turned into $n$ prediction tasks, one for each of its $n$ slots. Each prediction task is then performed by the trained model and predictions of high probability that differ from the existing variable in the corresponding slot are provided to the programmer as likely VARMISUSE bugs.

Unfortunately, this enumerative strategy has some key technical drawbacks. First, it approximates the repair process for a given program by enumerating over a number of independent prediction problems, where important shared context among the dependent predictions is lost. Second, in the training process, the synthetic bug is always only at the position of the slot. If for example, the program in Figure 1b were used for training, then five training examples, one corresponding to each identifier in a blue box (a variable read, in this case), would be generated. In each of them, the synthetic bug is exactly at the slot position. However, during inference, the model generates one prediction problem for each variable use in the program. In only one of these prediction problems does the slot coincide with the bug location; in the rest, the model now encounters a situation where there is a bug somewhere else, *at a location other than the slot*. This differs from the cases it has been trained on. For example, in Figure 1a, the prediction problem corresponding to the slot on line 5 contains a bug elsewhere (at line 6) and not in the slot. Only the problem corresponding to the slot on line 6 would match how the model was trained. This mismatch between training and test distributions hampers the prediction accuracy of the model. In our experiments, it leads to an accuracy drop of $4\%$ to $14\%$, even in the non-enumerative setting, i.e., when the exact location of the bug is provided. Since the enumerative approach uses the prediction of the same variable as the original variable for declaring no bugs at that location, this phenomenon contributes to its worse performance. Another drawback of the enumerative approach is that it produces one prediction per slot in a program, rather than one prediction per program. Allamanis et al. (2018) deal with this by manually selecting a numerical threshold and reporting a bug (and its repair) only if the predicted probability for a repair is higher than that threshold. Setting a suitable threshold is difficult: too low a threshold can increase false positives and too high a threshold can cause false negatives.

In order to deal with these drawbacks, we present a model that jointly learns to perform: 1) classification of the program as either faulty or correct (with respect to VARMISUSE bugs), 2) localization of the bug when the program is classified as faulty, and 3) repair of the localized bug. One of the key insights of our joint model is the observation that, in a program containing a single VARMISUSE bug, a variable token can only be one of the following: 1) a buggy variable (the *faulty* location), 2) some occurrence of the correct variable that should be copied over the incorrect variable into the faulty location (a *repair* location), or 3) neither the faulty location nor a repair location. This arises from the fact that the variable in the fault location cannot contribute to the repair of any other variable – there is only one fault location – and a variable in a repair location cannot be buggy at the same time. This observation leads us to a pointer model that can point at locations in the input (Vinyals et al., 2015) by learning distributions over input tokens. The hypothesis that a program that contains a bug at a location likely contains ingredients of the repair elsewhere in the program (Engler et al., 2001) has been used quite effectively in practice (Le Goues et al., 2012). Mechanisms based on pointer networks can play a useful role to exploit this observation for repairing programs.

We formulate the problem of classification as pointing to a special *no-fault* location in the program. To solve the joint prediction problem of classification, localization, and repair, we lift the usual pointer-network architecture to multi-headed pointer networks, where one pointer head points to the faulty location (including the no-fault location when the program is predicted to be non-faulty) and another to the repair location. We compare our joint prediction model to an enumerative approach for repair. Our results show that the joint model not only achieves a higher classification, localization, and repair accuracy, but also results in high true positive score.

Furthermore, we study how a pointer network on top of a recurrent neural network compares to the graph neural network used previously by Allamanis et al. (2018). The comparison is performed for program repair given an a priori known bug location, the very same task used by that work. Limited to only syntactic inputs, our model outperforms the graph-based one by 7 percentage points. Although encouraging, this comparison is only limited to syntactic inputs; in contrast, the graph model uses both syntax and semantics to achieve state-of-the-art repair accuracy. In future work we plan to study how jointly predicting bug location and repair might improve the graph model when

bug location is unknown, as well as how our pointer-network-based model compares to the graph-based one when given semantics, in addition to syntax; the latter is particularly interesting, given the relatively simpler model architecture compared to message-passing networks (Gilmer et al., 2017).

In summary, this paper makes the following key contributions: 1) it presents a solution to the general variable-misuse problem in which enumerative search is replaced by a neural network that jointly localizes and repairs faults; 2) it shows that pointer networks over program tokens provide a suitable framework for solving the VARMISUSE problem; and 3) it presents extensive experimental evaluation over multiple large datasets of programs to empirically validate the claims.

## 2 RELATED WORK

Allamanis et al. (2018) proposed an enumerative approach for solving the VARMISUSE problem by making individual predictions for each variable use in a program and reporting back all variable discrepancies above a threshold, using a graph neural network on syntactic and semantic information. We contrast this paper to that work at length in the previous section.

Devlin et al. (2017b) propose a neural model for semantic code repair where one of the classes of bugs they consider is VARREPLACE, which is similar to the VARMISUSE problem. This model also performs an enumerative search as it predicts repairs for all program locations and then computes a scoring of the repairs to select the best one. As a result, it also suffers from a similar training/test data mismatch issue as Allamanis et al. (2018). Similar to us, they use a pooled pointer model to perform the repair task. However, our model uses multi-headed pointers to perform classification, localization, and repair jointly.

DeepFix (Gupta et al., 2017) and SynFix (Bhatia et al., 2018) repair syntax errors in programs using neural program representations. DeepFix uses an attention-based sequence-to-sequence model to first localize the syntax errors in a C program, and then generates a replacement line as the repair. SynFix uses a Python compiler to identify error locations and then performs a constraint-based search to compute repairs. In contrast, we use pointer networks to perform a fine-grained localization to a particular variable use, and to compute the repair. Additionally, we tackle variable misuses, which are semantic bugs, whereas those systems fix only syntax errors in programs.

The DeepBugs (Pradel & Sen, 2018) paper presents a learning-based approach to identifying name-based bugs. The main idea is to represent program expressions using a small set of features (e.g., identifier names, types, operators) and then compute their vector representations by concatenating the individual feature embeddings. By injecting synthetic bugs, the classifier is trained to predict program expressions as buggy or not for three classes of bugs: swapped function arguments, wrong binary operator, and wrong operand in a binary operation. Similar to previous approaches, it is also an instance of an enumerative approach. Unlike DeepBugs, which embeds a single expression, our model embeds the full input program (up to a maximum prefix length) and performs both localization and repair of the VARMISUSE bugs in addition to the classification task. Moreover, our model implements a pointer mechanism for representing repairs that often requires pointing to variable uses in other parts of the program that are not present in the same buggy expression.

Sk_p (Pu et al., 2016) is another enumerative neural program repair approach to repair student programs using an encoder-decoder architecture. Given a program, for each program statement $s_i$, the decoder generates a statement $s_i'$ conditioned on an encoding of the preceding statement $s_{i-1}$ and the following statement $s_{i+1}$. Unlike our approach, which can generate VARMISUSE repairs using a pointer mechanism, the Sk_p model would need to predict full program statements for repairing such bugs. Moreover, similar to the DeepBugs approach, it would be difficult for the model to predict repairs that include variables defined two or more lines above the buggy variable location.

Automated program repair (APR) has been an area of active research in software engineering. The traditional APR approaches (Gazzola et al., 2018; Monperrus, 2018; Motwani et al., 2018) differ from our work in the following ways: 1) They require a form of specification of correctness to repair a buggy program, usually as a logical formula/assertion, a set of tests, or a reference implementation; 2) They depend on hand-designed search techniques for localization and repair; 3) The techniques are applied to programs that violate their specifications (e.g., a program that fails some tests), which means that the programs are already known to contain bugs. In contrast, a recent line of research in APR is based on end-to-end learning, of which ours is an instance. Our solution (like some other

learning-based repair solutions) has the following contrasting features: 1) It does not require any specification of correctness, but learns instead to fix a common class of errors directly from source-code examples; 2) It does not perform enumerative search for localization or repair—we train a neural network to perform localization and repair directly; 3) It is capable of first classifying whether a program has the specific type of bug or not, and subsequently localizing and repairing it. The APR community has also designed some repair bechmarks, such as ManyBugs and IntroClass (Goues et al., 2015), and Defects4J (Just et al., 2014), for test-based program repair techniques. The bugs in these benchmarks relate to the expected specification of individual programs (captured through test cases) and the nature of bugs vary from program to program. These benchmarks are therefore suitable to evaluate repair techniques guided by test executions. Learning-based solutions like ours focus on common error types, so it is possible for a model to generalize across programs, and work directly on embeddings of source code.

## 3  Pointer Models for Localization and Repair of VarMisuse

We use pointer-network models to perform joint prediction of both the location and the repair for VarMisuse bugs. We exploit the property of VarMisuse that both the bug and the repair variable must exist in the original program.

The model first uses an LSTM (Hochreiter & Schmidhuber, 1997) recurrent neural network as an encoder of the input program tokens. The encoder states are then used to train two pointers: the first pointer corresponds to the location of the bug, and the second pointer corresponds to the location of the repair variable. The pointers are essentially distributions over program tokens. The model is trained end-to-end using a dataset consisting of programs assumed to be correct. From these programs, we create both synthetic buggy examples, in which a variable use is replaced with an incorrect variable, and bug-free examples, in which the program is used as is. For a buggy training example, we capture the location of the bug and other locations where the original variable is used as the labels for the two pointers. For a bug-free training example, the location pointer is trained to point to a special, otherwise unused no-fault token location in the original program. In this paper, we focus on learning to localize a single VarMisuse bug, although the model can naturally generalize to finding and repairing more bugs than one, by adding more pointer heads.

### 3.1  Problem Definition

We first define our extension to the VarMisuse problem, which we call the VarMisuseRepair problem. We define the problem with respect to a whole program's source code, although it can be defined for different program scopes: functions, loop bodies, etc. We consider a program $f$ as a sequence of tokens $f = \langle t_1, t_2, \cdots, t_n \rangle$, where tokens come from a vocabulary $\mathbb{T}$, and $n$ is the number of tokens in $f$. The token vocabulary $\mathbb{T}$ consists of both keywords and identifiers. Let $\mathbb{V} \subset \mathbb{T}$ denote the set of all tokens that correspond to variables (uses, definitions, function arguments, etc.). For a program $f$, we define $V_{\mathsf{def}}^f \subseteq \mathbb{V}$ as the set of all variables defined in $f$, including function arguments; this is the set of all variables that can be used within the scope, including as repairs for a putative VarMisuse bug. Let $V_{\mathsf{use}}^f \subseteq \mathbb{V} \times \mathbb{L}$ denote the set of all (token, location) pairs corresponding to variable uses, where $\mathbb{L}$ denotes the set of all program locations.

Given a program $f$, the goal in the VarMisuseRepair problem is to either predict if the program is already correct (i.e., contains no VarMisuse bug) or to identify two tokens: 1) the location token $(t_i, l_i) \in V_{\mathsf{use}}^f$, and 2) a repair token $t_j \in V_{\mathsf{def}}^f$. The location token corresponds to the location of the VarMisuse bug, whereas the repair token corresponds to any occurrence of the correct variable in the original program (e.g., its definition or one of its other uses), as illustrated in Figure 2a. In the example from Figure 2a, $\mathbb{T}$ contains all tokens, including variables, literals, keywords; $\mathbb{V}$ contains `sources`, `object_name`, `subject_name`, and `result`; $V_{\mathsf{use}}^f$ contains the uses of variables, e.g., `sources` at its locations on lines 2 and 3, and $V_{\mathsf{def}}^f$ is the same as $\mathbb{V}$ in this example.[1]

---

[1]Note that by a variable use, we mean the occurrence of a variable in a load context. However, this definition of variable use is arbitrary and orthogonal to the model. In fact, we use the broader definition of *any* variable use, load or store, when comparing to Allamanis et al. (2018), to match their definition and a fair comparison to their results (see Section 4).

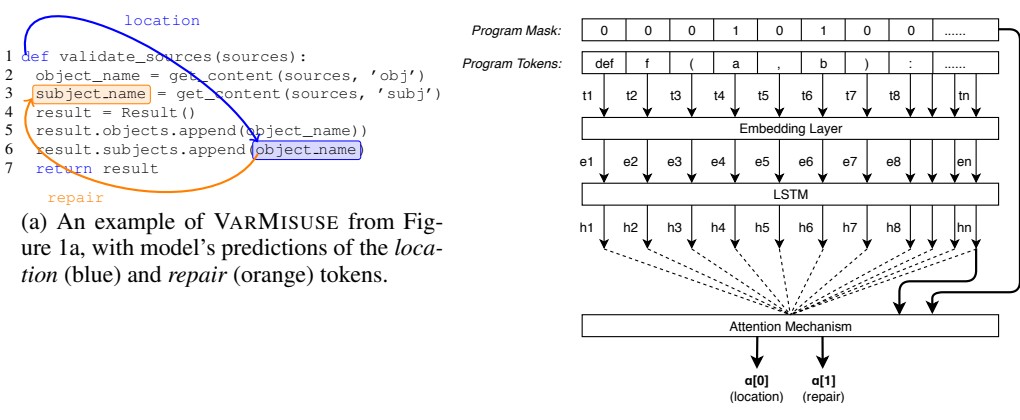

(a) An example of VARMISUSE from Figure 1a, with model's predictions of the *location* (blue) and *repair* (orange) tokens.

(b) Design details of our model.

Figure 2: Illustration of our model on a real-example (left), and design of the model (right).

## 3.2 MULTI-HEADED POINTER MODELS

We now define our pointer network model (see Figure 2b).

Given a program token sequence $f = \langle t_1, t_2, \cdots, t_n \rangle$, we embed the tokens $t_i$ using a trainable embedding matrix $\phi : \mathbb{T} \rightarrow \mathbb{R}^d$, where $d$ is the embedding dimension. We then run an LSTM over the token sequence to obtain hidden states (dimension $h$) for each embedded program token.

$$[e_1, \cdots, e_n] = [\phi(t_1), \cdots, \phi(t_n)] \tag{1}$$
$$[h_1, \cdots, h_n] = \mathsf{LSTM}(e_1, \cdots, e_n) \tag{2}$$

Let $m \in \{0, 1\}^n$ be a binary vector such that $m[i] = 1$ if the token $t_i \in V_{\mathsf{def}}^f$ or $(t_i, \_) \in V_{\mathsf{use}}^f$, otherwise $m[i] = 0$. The vector $m$ acts as a masking vector to only consider hidden states that correspond to states of the variable tokens. Let $\mathbf{H} \in \mathbb{R}^{h \times n}$ denote a matrix consisting of the hidden-state vectors of the LSTM obtained after masking, i.e., $\mathbf{H} = m \odot [h_1 \cdots h_n]$. We then perform attention over the hidden states using a mechanism similar to that of Rocktäschel et al. (2016) as follows:

$$\mathbf{M} = \tanh(\mathbf{W_1}\mathbf{H} + \mathbf{W_2}h_n \otimes \mathbf{1}_n) \qquad \mathbf{M} \in \mathbb{R}^{h \times n} \tag{3}$$

where $\mathbf{W_1}, \mathbf{W_2} \in \mathbb{R}^{h \times h}$ are trainable projection matrices and $\mathbf{1}_n \in \mathbb{R}^n$ is a vector of ones used to obtain $n$ copies of the final hidden state $h_n$.

We then use another trained projection matrix $\mathbf{W} \in \mathbb{R}^{h \times 2}$ to compute the $\alpha \in \mathbb{R}^{2 \times n}$ as follows:

$$\alpha = \mathsf{softmax}(\mathbf{W}^\intercal \mathbf{M}) \qquad \alpha \in \mathbb{R}^{2 \times n} \tag{4}$$

The attention matrix $\alpha$ corresponds to two probability distributions over the program tokens. The first distribution $\alpha^\intercal[0] \in \mathbb{R}^n$ corresponds to location token indices and the second distribution $\alpha^\intercal[1] \in \mathbb{R}^n$ corresponds to repair token indices. We experiment with and without using the masking vector on the hidden states, and also using masking on the unnormalized attention values.

## 3.3 TRAINING THE MODEL

We train the pointer model on a synthetically generated training corpus consisting of both buggy and non-buggy Python programs. Starting from a publicly available dataset of Python files, we construct the training, validation, and evaluation datasets in the following manner. We first collect the source code for each program definition from the Python source files. For each program definition $f$, we collect the set of all variable definitions $V_{\mathsf{def}}^f$ and variable uses $V_{\mathsf{use}}^f$. For each variable use $(v_u, l_u) \in V_{\mathsf{use}}^f$, we replace its occurrence by another variable $v_d \in V_{\mathsf{def}}^f$ to obtain an example

ensuring the following conditions: 1) $v_d \neq v_u$, 2) $|V_{\text{def}}^f| > 1$, i.e., there are at least 2 possible variables that could fill the slot at $l_u$.

Let $i$ denote the token index in the program $f$ of the variable $v_u$ chosen to be replaced by another (incorrect) variable $v_d$ in the original program. We then create two binary vectors $\text{Loc} \in \{0,1\}^n$ and $\text{Rep} \in \{0,1\}^n$ in the following manner:

$$\text{Loc}[m] = \begin{cases} 1, & \text{if } i = m \\ 0, & \text{otherwise} \end{cases} \qquad\qquad \text{Rep}[m] = \begin{cases} 1, & \text{if } v_u = t_m \\ 0, & \text{otherwise} \end{cases}$$

$\text{Loc}$ is a *location* vector of length $n$ (program length) which is 1 at the location containing bug, otherwise 0. $\text{Rep}$ is a *repair* vector of length $n$ which is 1 at all locations containing the variable $v_u$ (correct variable for the location $i$), otherwise 0.

For each buggy training example in our dataset, we also construct a non-buggy example where the replacement is not performed. This is done to obtain a 50-50 balance in our training datasets for buggy and non-buggy programs. For non-buggy programs, the target location vector $\text{Loc}$ has a special token index 0 set to 1, i.e. $\text{Loc}[0] = 1$, and the value at all other indices is 0.

We use the following loss functions for training the location and repair pointer distributions.

$$L_{\text{loc}} = -\sum_{i=1}^{n}(\text{Loc}[i] \times \log(\alpha^\intercal[0][i])) \qquad\qquad L_{\text{rep}} = -\sum_{i=1}^{n}(\text{Rep}[i] \times \log(\alpha^\intercal[1][i]))$$

The loss function for the repair distribution adds up the probabilities of target pointer locations. We also experiment with an alternative loss function $L'_{\text{rep}}$ that computes the maximum of the probabilities of the repair pointers instead of their addition.

$$L'_{\text{rep}} = -\max(\text{Rep}[i] \times \log(\alpha^\intercal[1][i]))$$

The joint model optimizes the additive loss $L_{\text{joint}} = L_{\text{loc}} + L_{\text{rep}}$. The enumerative solution discussed earlier forms a baseline method. We specialize the multi-headed pointer model to produce only the repair pointer and use it within the enumerative solution for predicting repairs.

## 4    EVALUATION

In our experimental evaluation, we evaluate three research questions. First, is the joint prediction model VARMISUSEREPAIR effective in finding VARMISUSE bugs in programs and how does it compare against the enumerative solution (Section 4.1)? Second, how does the presence of as-yet-unknown bugs in a program affect the bug-finding effectiveness of the VARMISUSE repair model even in the non-enumerative case (Section 4.2)? Third, how does the repair pointer model compare with the graph-based repair model by Allamanis et al. (2018) (Section 4.3)?

**Benchmarks**    We use two datasets for our experiments. Primarily, we use ETH-Py150[2], a public corpus of GitHub Python files extensively used in the literature (Raychev et al., 2016; Vechev & Yahav, 2016). It consists of 150K Python source files, already partitioned by its publishers into training and test subsets containing 100K and 50K files, respectively. We split the training set into two sets: training (90K) and validation (10K). We further process each dataset partition by extracting unique top-level functions, resulting in 394K (training), 42K (validation), and 214K (test) unique functions. For each function, we identify VARMISUSE slots and repair candidates. For each function and slot pair, we generate one bug-free example (without any modification) and one buggy example by replacing the original variable at the slot location by a randomly chosen incorrect variable. More details about the data generation is presented in the appendix (Section A). Because of the quadratic complexity of evaluating the enumerative model, we create a smaller evaluation set by sampling 1K

---

[2]https://www.sri.inf.ethz.ch/py150

test files that results in 12,218 test examples (out of which half are bug-free). For the evaluation set, we construct a bug-free and buggy-example per function using the procedure defined before, but now by also randomly selecting a single slot location in the function instead of creating an example for each location for the training dataset. All our evaluation results on the ETH dataset use this filtered evaluation set. Note that the inputs to the enumerative model and the joint model are different; the joint model accepts a complete program, while the enumerative model accepts a program with a hole that identifies a slot. For this reason, training, validation, and test datasets for the enumerative approach are constructed by inserting a hole at variable-use locations.

Our second dataset, MSR-VarMisuse, is the public portion of the dataset used by Allamanis et al. (2018). It consists of 25 C# GitHub projects, split into four partitions: train, validation, *seen* test, and *unseen* test, consisting of 3738, 677, 1807, and 1185 files each. The seen test partition contains (different) files from the same projects that appear in the train and validation partitions, whereas the unseen test partition contains entire projects that are disjoint from those in test and validation.

Note the differences between the two datasets: ETH-Py150 contains Python examples with a function-level scope, slots are variable loads, and candidates are variables in the scope of the slot (Python is dynamically typed, so no type information is used); in contrast, MSR-VarMisuse contains C# examples that are entire files, slots are both load and store uses of variables, and repair candidates are all variables in the slot's scope with an additional constraint that they are also type-compatible with the slot. We use the ETH-Py150 dataset for most of our experiments because we are targeting Python, and we use MSR-VarMisuse when comparing to the results of Allamanis et al. (2018). The average number of candidates per slot in the ETH-Py150 dataset is about 9.26, while in MSR-VarMisuse it is about 3.76.

## 4.1 Joint Model vs. Enumerative Approach

We first compare the accuracy of the joint model (Section 3.2) to that of an enumerative repair model, similar in spirit (but not in model architecture) to that by Allamanis et al. (2018). For the enumerative approach, we first train a pointer network model $M_r$ to only predict repairs for a given program and slot. At test time, given a program $P$, the enumerative approach first creates $n$ variants of $P$, one per slot. We then use the trained model $M_r$ to predict repairs for each of the $n$ variants and combine them into a single set. We go through the predictions in decreasing order of probability, until a prediction modifies the original program. If no modifications happen, then it means that the model classifies the program under test as a bug-free program. We define two parameters to filter the predictions: 1) $\tau$: a threshold value for probabilities to decide whether to return the predictions, and 2) $k$: the maximum number of predictions the enumerative approach is allowed to make.

The results for the comparison for different $\tau$ and $k$ values are shown in Table 1. We measure the following metrics: 1) **True Positive**, the percentage of the bug-free programs in the ground truth classified as bug free; 2) **Classification Accuracy**, the percentage of total programs in the test set classified correctly as either bug free or buggy; 3) **Localization Accuracy**, the percentage of buggy programs for which the bug location is correctly predicted by the model; and 4) **Localization+Repair Accuracy**, the percentage of buggy programs for which both the location and repair are correctly predicted by the model.

The table lists results in decreasing order of prediction permissiveness. A higher $\tau$ value (and lower $k$ value) reduces the number of model predictions compared to lower $\tau$ values (and higher $k$ values). As expected, higher $\tau$ and lower $k$ values enable the enumerative approach to achieve a higher true positive rate, but a lower classification accuracy rate. More importantly for buggy programs, the localization and repair accuracy drop quite sharply. With lower $\tau$ and higher $k$ values, the true positive rate drops dramatically, while the localization and repair accuracy improve significantly. In contrast, our joint model achieves a maximum localization accuracy of 71% and localization+repair accuracy of 65.7%, an improvement of about 6.4% in localization and about 9.9% in localization+repair accuracy, compared to the lowest threshold and highest $k$ values. Remarkably, the joint model achieves such high accuracy while maintaining a high true-positive rate of 84.5% and a high classification accuracy of 82.4%. This shows that the network is able to perform the localization and repair tasks jointly, efficiently, and effectively, without the need of an explicit enumeration.

| Model | True Positive | Classification Accuracy | Localization Accuracy | Localization+Repair Accuracy |
|---|---|---|---|---|
| **Enumerative** | | | | |
| Threshold ($\tau = 0.99$) | **99.9**% | 53.5% | 7.0% | 7.0% |
| Threshold ($\tau = 0.9$) | 99.7% | 56.7% | 13.4% | 13.3% |
| Threshold ($\tau = 0.7$) | 99.0% | 59.2% | 18.3% | 17.9% |
| Threshold ($\tau = 0.5$) | 95.3% | 63.8% | 28.7% | 27.1% |
| Threshold ($\tau = 0.3$) | 81.1% | 68.6% | 44.2% | 39.7% |
| Threshold ($\tau = 0.2$) | 66.3% | 70.6% | 54.3% | 47.4% |
| Threshold ($\tau = 0$) | 42.2% | 71.1% | 64.6% | 55.8% |
| Top-k ($k = 1$) | 91.7% | 63.6% | 27.2% | 24.8% |
| Top-k ($k = 3$) | 64.9% | 70.1% | 49.6% | 43.2% |
| Top-k ($k = 5$) | 50.9% | 70.9% | 58.4% | 50.4% |
| Top-k ($k = 10$) | 43.5% | 71.1% | 63.6% | 54.8% |
| Top-k ($k = \infty$) | 42.2% | 71.1% | 64.6% | 55.8% |
| **Joint** | 84.5% | **82.4%** | **71%** | **65.7%** |

Table 1: The overall evaluation results for the joint model vs. the enumerative approach (with different threshold $\tau$ and top-k $k$ values) on the `ETH-Py150` dataset. The enumerative approach uses a pointer network model trained for repair.

**Performance Comparison:** In addition to getting better accuracy results, the joint model is also more efficient for training and prediction tasks. During training, the examples for the pointer model are easier to batch compared to the GGNN model in Allamanis et al. (2018) as different programs lead to different graph structures. Moreover, as discussed earlier, the enumerative approaches require making $O(n)$ predictions at inference time, where $n$ denotes the number of variable-use locations in a program. On the other hand, the joint model only performs a single prediction for the two pointers given a program.

## 4.2 Effect of Incorrect Slot Placement

We now turn to quantifying the effect of incorrect slot placement, which occurs frequently in the enumerative approach: $n - 1$ out of $n$ times for a program with $n$ slots. We use the same repair-only model from Section 4.1, but instead of constructing an enumerative bug localization and repair procedure out of it, we just look at a single repair prediction.

We apply this repair-only model to a test dataset in which, in addition to creating a prediction problem for a slot, we also randomly select one other variable use in the program (other than the slot) and replace its variable with an incorrect in-scope variable, thereby introducing a VARMISUSE bug away from the slot of the prediction problem. We generate two datasets: AddBugAny, in which the injected VARMISUSE bug is at a random location, and AddBugNear, in which the injection happens within two variable-use locations from the slot, and in the first 30 program tokens; we consider the latter a tougher, more adversarial case for this experiment. The corresponding bug-free datasets are NoBugAny and NoBugNear with the latter being a subset of the former. We refer to two experiments below: Any (comparison between NoBugAny and AddBugAny) and Near (comparison between NoBugNear and AddBugNear).

Figure 3 shows our results. Figure 3a shows that for Any, the model loses significant accuracy, dropping about $4.3$ percentage points for $\tau = 0.5$. The accuracy drop is lower as a higher prediction probability is required by higher $\tau$, but it is already catastrophically low. Results are even worse for the more adversarial Near. As shown in Figure 3b, accuracy drops between $8$ and $14.6$ percentage points for different reporting thresholds $\tau$.

These experiments show that a repair prediction performed on an unlikely fault location can significantly impair repair, and hence the overall enumerative approach, since it relies on repair predictions for both localization and repair. Figure 4 shows some repair predictions in the presence of bugs.

| Threshold | Repair Accuracy | | Accuracy |
| Value | NoBugAny | AddBugAny | Drop |
| --- | --- | --- | --- |
| $\tau = 0$ | 80.8% | 76.2% | 4.6% |
| $\tau = 0.2$ | 60.0% | 55.5% | 4.5% |
| $\tau = 0.3$ | 40.8% | 36.6% | 4.2% |
| $\tau = 0.5$ | 19.1% | 14.8% | 4.3% |
| $\tau = 0.7$ | 8.5% | 5.5% | 3.0% |
| $\tau = 0.9$ | 5.5% | 2.5% | 3.0% |
| $\tau = 0.99$ | 2.4% | 1.0% | 1.4% |

(a) Testing of repair-only model on Any.

| Threshold | Repair Accuracy | | Accuracy |
| Value | NoBugNear | AddBugNear | Drop |
| --- | --- | --- | --- |
| $\tau = 0$ | 88.6% | 80.2% | 8.4% |
| $\tau = 0.2$ | 81.5% | 73.2% | 8.3% |
| $\tau = 0.3$ | 68.7% | 59.9% | 8.8% |
| $\tau = 0.5$ | 44.6% | 30.0% | 14.6% |
| $\tau = 0.7$ | 24.1% | 13.0% | 11.1% |
| $\tau = 0.9$ | 18.1% | 6.9% | 11.2% |
| $\tau = 0.99$ | 8.5% | 2.7% | 5.8% |

(b) Testing of repair-only model on Near.

Figure 3: The drop in repair accuracy of the repair model due to incorrect slot placement.

```
1 def GetFriends(self, user, page):
2   if not user and not self._username:
3     raise TwitterError('twitter.Api')
4 if user:
5   url = 'http://%s.json' % user
6 else:
7   url = 'http://s.json'
8 parameters = {}
9 if <slot>:
10   parameters['page'] = page
11 json = self._FetchUrl(url, parameters=parameters)
12 data = simplejson.loads(json)
```

Prediction: page (33.7%) ✓

```
1 def GetFriends(self, user, page):
2   if not user and not self._username:
3     raise TwitterError('twitter.Api')
4 if user:
5   url = 'http://%s.json' % user
6 else:
7   url = 'http://s.json'
8 parameters = {}
9 if <slot>:
10   parameters['page'] = data
11 json = self._FetchUrl(url, parameters=parameters)
12 data = simplejson.loads(json)
```

Prediction: data (39.5%) ✗

```
1 def multisock(self,name,_type):
2   if _type:
3     return MultiSocket(name, <slot>)
4   else:
5     return MultiSocket(name)
```

Prediction: _type (66.92%) ✓

```
1 def multisock(self,name,_type):
2   if _type:
3     return MultiSocket(_type, <slot>)
4   else:
5     return MultiSocket(name)
```

Prediction: name (35.8%) ✗

```
1 def closable(self):
2   item = self.dockItem()
3   if item is not None:
4     return <slot>.closable()
5   return True
```

Prediction: item (47.2%) ✓

```
1 def closable(self):
2   item = self.dockItem()
3   if self is not None:
4     return <slot>.closable()
5   return True
```

Prediction: self (32.2%) ✗

Figure 4: A sample of differing repair predictions (and prediction probabilities) for slots shown in blue and injected bugs enclosed in red.

### 4.3 COMPARISON OF GRAPH AND POINTER NETWORKS

We now compare the repair-only model on MSR-VarMisuse, the dataset used by the state-of-the-art VARMISUSE localization and repair model by Allamanis et al. (2018). Our approach deviates in three primary ways from that earlier one: 1) it uses a pointer network on top of an RNN encoder rather than a graph neural network, 2) it does separate but joint bug localization and repair rather than using repair-only enumeratively to solve the same task, and 3) it applies to syntactic program information only rather than syntax and semantics. Allamanis et al. (2018) reported in their ablation study that their system, on syntax only, achieved test accuracy of $55.3\%$ on the "seen" test; on the same test data we achieve $62.3\%$ accuracy. Note that although the test data is identical, we trained on the published training dataset[3], which is a subset of the unpublished dataset used in that ablation study. We get better results even though our training dataset is about 30% smaller than their dataset.

### 4.4 EVALUATION ON VARIABLE MISUSE IN PRACTICE

In order to evaluate the model on realistic scenarios, we collected a dataset from multiple software projects in an industrial setting. In particular, we identified pairs of consecutive snapshots of functions from development histories that differ by a single variable use. Such before-after pairs

---

[3]https://aka.ms/iclr18-prog-graphs-dataset

| Model | True Positive | Classification Accuracy | Localization Accuracy | Localization+Repair Accuracy |
|---|---|---|---|---|
| Joint | **67.3%** | **66.7%** | **21.9%** | **15.8%** |
| Enumerative | 41.7% | 47.2% | 6.1% | 4.5% |

Table 2: The comparison of the joint model vs the enumerative approach on programs collected in an industrial setting.

of function versions indicate likely variable misuses, and several instances of them were explicitly marked as VARMISUSE bugs by code reviewers during the manual code review process.

More precisely, we find two snapshots $f$ and $f'$ of the same program for which $V_{\text{def}}^{f} = V_{\text{def}}^{f'}$, $V_{\text{use}}^{f} = V_{\text{common}}^{f} \cup (t_i, l_i)$, and $V_{\text{use}}^{f'} = V_{\text{common}}^{f} \cup (t'_i, l_i)$, where $t_i \neq t'_i$, and $t_i, t'_i \in V_{\text{def}}^{f}$. For each such before-after snapshot pair $(f, f')$, we collected all functions from the same file in which $f$ was present. We expect our model to classify all functions other than $f$ as bug-free. For the function $f$, we want the model to classify it as buggy, and moreover, localize the bug at $l_i$, and repair by pointing out token $t'_i$. In all, we collected $4592$ snapshot pairs. From these, we generated a test dataset of $41672$ non-buggy examples and $4592$ buggy examples. We trained the pointer model on a training dataset from which we exclude the $4592$ files containing the buggy snapshots. The results of the joint model and the best localization and repair accuracies achieved by the enumerative baseline approach are shown in Table 2. The joint model achieved a true positive rate of $67.3\%$, classification accuracy of $66.7\%$, localization accuracy of $21.9\%$ and localization+repair accuracy of $15.8\%$. These are promising results on data collected from real developer histories and in aggregate, our joint model could localize and repair 727 variable misuse instances on this dataset. On the other hand, the enumerative approach achieved significantly lower values of true positive rate of $41.7\%$, classification accuracy of $47.2\%$, localization accuracy of $6.1\%$, and localization+repair accuracy of $4.5\%$.

## 5 CONCLUSION

In this paper, we present an approach that jointly learns to localize and repair bugs. We use a key insight of the VARMISUSE problem that both the bug and repair must exist in the original program to design a multi-headed pointer model over a sequential encoding of program token sequences. The joint model is shown to significantly outperform an enumerative approach using a model that can predict a repair given a potential bug location. In the future, we want to explore joint localization and repair using other models such as graph models and combinations of pointer and graph models, possibly with using more semantic information about programs.

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

## A  TRAINING DATA GENERATION

For each Python function in the `ETH-Py150` dataset, we identify VARMISUSE slots and repair candidates. We choose as slots only uses of variables in a load context; this includes explicit reads from variables in right-hand side expressions (`a = `x` + y`), uses as function-call arguments (`func(`x`, y)`), indices into dictionaries and lists even on left-hand side expressions (`sequence[`x`] = 13`), etc. We define as repair candidates all variables that are in the scope of a slot, either defined locally, imported globally (with the Python `global` keyword), or as formal arguments to the enclosing function.

For each slot in a function (variable use locations), we generate one buggy example, as long as there are at least two repair candidates for a slot (otherwise, the repair problem would be trivially solved by picking the only eligible candidate); we discard slots and corresponding examples with only trivial repair solutions, and we discard functions and corresponding examples with only trivial slots. For each buggy example, we also generate one bug-free example, by leaving the function as is and marking it (by assumption) as correct. Note that this results in duplicate copies of correct functions in the training dataset. To illustrate, using the correct function in Figure 1b, we would generate five bug-free examples (labeling the function, as is, as correct), and one buggy example per underlined slot (inserting an incorrect variable chosen at random), each identifying the current variable in the slot as the correct repair, and the variables `sources`, `object_name`, `subject_name`, and `result` as repair candidates. Although buggy repair examples are defined in terms of candidate variable *names*, any mention of a candidate in the program tokens can be pointed to by the pointer model; for example, the repair pointer head, when asked to predict a repair for the slot on line 6, could point to the (incorrect) variable `sources` appearing on lines 1, 2, or 3, and we don't distinguish among those mentions of a predicted repair variable when it is the correct prediction.

The `MSR-VarMisuse` dataset consists of 25 C# GitHub projects, split into four partitions: train, validation, *seen* test, and *unseen* test, consisting of 3738, 677, 1807, and 1185 files each. The seen test partition contains (different) files from the same projects that appear in the train and validation partitions, whereas the unseen test partition contains entire projects that are disjoint from those in test and validation. The published dataset contains pre-tokenized token sequences, as well as VARMISUSE repair examples, one per slot present in every file, as well as associated repair candidates for that slot, and the correct variable for it. This dataset defines slots as variable uses in both load and store contexts (e.g., even left-hand side expressions), and candidates are type-compatible with the slot. Every example has at least two repair candidates.

