# OpenReview forum: "Neural Program Repair by Jointly Learning to Localize and Repair"
_ICLR.cc/2019/Conference_

### Official Review · AnonReviewer3 · 2018-11-02
**Weak baseline and lack trade-offs discussion makes it hard to say if idea is good.**

**Rating:** 5
**Confidence:** 5

**Review:**

Several recent works propose to discover bugs in code by creating dataset of presumably correct code and then to augment the data by introducing a bug and creating a classifier that would discriminate between the buggy and the correct version. Then, this classifier would be used to predict at each location in a program if a bug is present.

This paper hypothetizes that when running on buggy code (to discover the bug) would lead to such classifier misbehave and report spurious bugs at many other locations besides the correct one and would fail at precisely localizing the bug. Then, they propose a solution that essentially create a different classifier that is trained to localize the bug.

Unfortunatley this leads to a number of weaknesses:
 - The implementation and evaluation are only on a quite syntactic system with low precision and that needs to sift through a huge amount of weak and irrelevant signals to make predictions.
 - The gap here is huge: the proposed system is only based on program syntax and gets 62.3% accuracy, but state-of-the-art has 85.5% (there is actually another recent technique [1] also with accuracy in the >80% range)
 - It is not clear that the entire discussed problem is orthogonal to the selection of such weak baselines to build the improvements on.
 - Trade-offs are not clear: is the proposed architecture slower to train and query than the baselines?

Strengths of the paper are:
 - Well-written and easy to follow and understand.
 - Evaluation on several datasets.
 - Interesting architecture for bug-localization if the idea really works.

[1] Michael Pradel, Koushik Sen. DeepBugs: a learning approach to name-based bug detection

---

> ### Author Response · Authors · 2018-11-06
> **Response to AnonReviewer3**
>
> Thanks for the review and constructive feedback. We believe there are a few major misunderstandings in the review and we would like to take this opportunity to clarify them. We will be happy to discuss them in more detail if more clarifications might be needed or there are more questions.
>
> We would first like to point out that ours is the first model that jointly learns to perform both localization and repair of the variable misuse bugs. It exploits the property of this particular class of variable misuse bugs -- both the location and repair corresponds to variable use locations in the program. Unlike Allamanis et al. 2018 that uses an enumerative approach to make a number of predictions for a program that is linear in number of variable uses, our model makes a single prediction using a two pointer based mechanism.
>
> Thanks for the pointer to the DeepBugs paper. Note that there are several differences of our work with the DeepBugs paper, which we explain below. We will add this to our revision as well.
> 1. DeepBugs learns a classifier over single expressions. It takes a single program expression as input (e.g. “2% i == 0”) and classifies it as positive or negative. On the other hand, in addition to classifying programs, our model learns to localize and also repair the bug using a two-headed pointer network.
> 2. Our model uses the full program (up to 250 number of tokens) for learning the vector representation. DeepBugs only looks at a single expression at a time.
> 3. Finally, the 80% accuracy number for DeepBugs is only for expression classification. It has no direct comparison with our model’s accuracy since it is a different problem (classifying a single expression as correct compared to analyzing a full program to identify bug location and the corresponding repair). Moreover, our pointer models also get to 82.4% classification accuracy for full programs (Table 1).
>
> Allamanis et al. only report the accuracy of repair-only model, where the model predicts a single variable at a time for each slot location in a program. Translating their 85.5% repair accuracy number to a number that corresponds to repairing the full program would lead to a very different result. In Table 1, we try to replicate a similar experiment and show that jointly learning the model leads to significant improvements without sacrificing true positive and classification accuracy. Moreover, Allamanis et al. 2018 perform a significant amount of program preprocessing including type inference, control flow, and data flow analysis to add different types of graph edges. Without such pre-processing, they achieve an accuracy of 55.3% on repair-only tasks (Section 4.3). In our work, we want our distributed representations to automatically learn good representations of programs without any manual feature engineering.
>
> Performance trade-off: In fact, our proposed architecture is significantly more scalable and easier to train. Since we are using sequence models to compute pointer attentions that are easier to batch over multiple examples, it is much more scalable to train compared to graph models that are difficult to batch because of different graph sizes. Our own graph implementation was significantly slower to train.
>
> In addition to that, it is also significantly faster at inference time, as it does not need to perform an O(n) number of model predictions, where n is the number of variable use locations in the program under test. For our model, it performs a single prediction, which is much faster.
>
> Please let us know if this helped clarify the questions and comments.

---

> > ### Comment · AnonReviewer3 · 2018-11-08
> > **Paper can be updated**
> >
> > Thank you for the response. Certainly, many of the issues discussed can be incorporated in the paper, not in comments. The task the discussed papers introduce is used in practice to do anomaly detection and then to report bugs. The bug localization of Allamanis et al has very high positive rate (they did not share it in their paper, but text implies it is a few bugs per hundreds of reports). Pradel and Sen however share accuracy on a non-synthetic task and it is around 50%. It does not yet look like that the result of this submission will translate to any better bug-finding technique in practice, but I am looking forward to see if the proposed technique is a good idea on a more realistic scenario.
> >
> > Also, this is not the first paper to propose both localization and fixing. The following work does it and their accuracy is lower, but on a more practical task:
> > Pu, Yewen, Karthik Narasimhan, Armando Solar-Lezama, and Regina Barzilay. “Sk_p: a Neural Program Corrector for MOOCs.”, OOPSLA 2016
> >
> > One possibility to improve the submission is to try the neural approach on their dataset and report state-of-the-art results.

---

> > > ### Author Response · Authors · 2018-11-12
> > > **Paper Updated**
> > >
> > > Thank you for more comments and helpful suggestions.
> > >
> > > Revised version of the paper is added. We incorporated some of the above discussion on comparison with DeepBugs (in section 2 - related work) and a discussion on the performance comparison (in section 4.1). We also added a discussion on differences with the Sk_p paper (in section 2), and the results of our model in realistic scenarios (section 4.4), which we also explain below.
> > >
> > > Thanks for the suggestion on evaluating the model on realistic scenarios. We have been collecting such a dataset for evaluating the model. In particular, we examined development histories in a software company (name elided for anonymity) to extract pairs of consecutive snapshots of code (on a level of functions) which differ by a single variable occurrence. These are indicative of variable misuses; several of which were explicitly pointed out as bugs by code reviewers during the manual code review process. For each snapshot pair (x,y), where x is a function before change and y is the same function after change, we collected all functions from the same file in which function x was present. We expect our model to classify all functions other than x as bug-free. For the function x, we want the model to classify it as buggy, and moreover, localize and repair the bug where the repair is the difference between y and x. In all, we collected 4592 such snapshot pairs. From these, we generated a test dataset of 41672 non-buggy examples and 4592 buggy examples. We trained the pointer model on a training dataset from which we exclude the 4592 files containing the buggy snapshots. When applied on the test dataset, the model achieved true positive rate of 67.3%, classification accuracy of 66.7%, localization accuracy of 21.9% and localization+repair accuracy of 15.8%. In all, on this dataset, our model could localize and repair 727 variable misuse instances. These are promising results on data collected from real developer histories. We have also added a subsection 4.4 to describe the evaluation and the results.
> > >
> > > Relationship with Sk_p and applying the model on their dataset:
> > > We would like to point out that Sk_p does not perform direct localization, instead it performs an enumerative search (for potential bug locations) which we eschew. Given a program, it considers each statement individually. For each program statement s_i, it considers the previous program statement s_{i-1} and the following statement s_{i+1} as inputs to an encoder while the decoder generates the full statement s’_i that should be present in between the two statements. It performs this prediction for each statement individually and then reports discrepancies as repairs. Note that it can only produce full statements as repairs unlike our approach for predicting a single variable usage repair. Moreover, similar to the DeepBugs approach, it would be difficult for the model to predict repairs that include variables present at two or more lines above/below the buggy variable location.
> > >
> > > Their dataset is unfortunately not useful for our evaluation for the following reasons:
> > > 1. Our model is designed for the variable misuse problem, which typically occurs when programmers copy-paste code fragments and forget to change certain variables. The sk_p dataset is coming from small student programs (5-10 LOC) submitted for MOOC exercises, which will likely not have many variable misuse bugs.
> > > 2. The Sk_p model (similar to the SynFix paper) exploits the fact that many students are solving the same programming problem and likely will write similar solutions which can then be used to train the models. In our case, we are generalizing from programs written by developers for different tasks.

---

> > > > ### Comment · AnonReviewer3 · 2018-11-19
> > > > **Baselines**
> > > >
> > > > It seems that the real dataset has a different distributions than the synthetic bugs dataset based on Py150. Did you observe similar improvements over the baselines on it?
> > > >
> > > > In general, all current bug-finding research suffers from having no sense of "recall" of whether it discovers only very few bugs or it discovers most of the bugs from a certain class. Since the paper aims to look into such issues, this would be good to say what is happening on real data and also why  10% of the samples were chosen as being buggy, while the frequency of the bug is likely much lower. And localization accuracy is in the 21.9% range - still low, do the anomaly-based baseline techniques get to similar numbers?

---

> > > > > ### Author Response · Authors · 2018-11-19
> > > > > **Response**
> > > > >
> > > > > We thank the reviewer for the suggestion to compare the results of the joint model on the real dataset (Section 4.4) with the enumerative baseline. The repair-only model (underlying the enumerative baseline) from Section 4.1 is trained on the Py150 dataset. For a fair comparison, we have started training the repair-only model on the dataset used for training the joint model in Section 4.4. We will report the results (analogous to Table 1) as soon as the model training is finished in a few days. However, note that the issues with enumerative approaches are fundamental and independent of the choice of datasets.
> > > > >
> > > > > In this paper, our aim is to develop new models that can better capture certain family of program repair tasks (such as VarMisuse) and improve upon previous enumerative approaches on datasets previously used in the literature.
> > > > >
> > > > > Further, in order to evaluate how well the model performs on realistic scenarios, we created the dataset used in Section 4.4. Note that the dataset was created by capturing pairs of consecutive snapshots of functions from development histories that differ by a single variable occurrence. We then also include other functions in those files that weren’t changed as sources of correct functions. We did not intentionally set the percentage of buggy samples to 10%, but it was an artifact of the way we created our dataset by including non-buggy functions from the files that differed by one variable change in consecutive snapshots. The rationale behind this procedure was that it estimates when given files with VarMisuse bugs, in how many cases can the model learn to classify the faulty functions (from among all functions in the file) as faulty and further localize and repair the bugs in those faulty functions.

---

> > > > > > ### Author Response · Authors · 2018-11-28
> > > > > > **Baseline results on the real dataset**
> > > > > >
> > > > > > We have finished evaluating the baseline enumerative model on the real dataset (Section 4.4). The best localization and localization+repair accuracies achieved by the enumerative approach are 6.1% and 4.5% respectively. The corresponding True positive and Classification Accuracy for the model are 41.7% and 47.2%. In contrast, our joint model achieves significantly higher accuracies (+15% for localization and +11% for localization+repair) as reported in Section 4.4 (also provided below for comparison).
> > > > > >
> > > > > >                                           True Positive     Classification    Localization    Localization+Repair
> > > > > > Joint Model			           67.3%		      66.7%	        21.9%	          15.8%
> > > > > > Enumerative Model                41.7%                47.2%                6.1%                  4.5%
> > > > > >
> > > > > > We will add this result to Section 4.4 as well.

---

### Official Review · AnonReviewer1 · 2018-11-13
**Simple model and really good results, but uninteresting contributions**

**Rating:** 7
**Confidence:** 4

**Review:**

This paper considers the problem of VarMisuse, a kind of software bug where a variable has been misused. Existing approaches to the problem create a complex model, followed by enumerating all possible variable replacements at all possible positions, in order to identify where the bug may exist. This can be problematic for training which is performed using synthetic replacements; enumeration on non-buggy positions does not reflect the test case. Also, at test time, enumerating is expensive, and does not accurately capture the various dependencies of the task. This paper instead proposes a LSTM based model with pointers to break the problem down into multiple steps: (1) is the program buggy, (2) where is the bug, and (3) what is the repair. They evaluate on two datasets, and achieve substantial gains over previous approaches, showing that the idea of localizing and repairing and effective.

I am quite conflicted about this paper. Overall, the paper has been strengths:
- It is quite well-written, and clear. They do a good job of describing the problems with earlier approaches, and how their approach can address it.
- The proposed model is straightforward, and addresses the problem quite directly. There is elegance in its simplicity.
- The evaluation is quite thorough, and the resulting gains are quite impressive.

However, I have some significant reservations about the novelty and the technical content. The proposed model doesn't quite bring anything new to the table. It is a straightforward combination of LSTMs with pointers, and it's likely the benefits are coming from the reformulation of the problem, not from the actual proposed model. This, along with the fact that VarMisuse is a small subset of the kinds of bugs that can appear in software, makes me feel the ideas in this paper may not lead to significant impact on the research community.

As a minor aside, this paper addresses some specific aspects of VarMisuse task and the Allamanis et al 2018 model, and introduces a model just for it. I consider the Allamanis model a much more general representation of programs, and much more applicable to other kinds of debugging tasks (but yes, since they didn't demonstrate this either, I'm not penalizing this paper for it).

--- Update ----
Given the author's response and the discussion, I'm going to raise the score a little. Although there are some valid concerns, it provides a clear improvement over Allamanis et al paper, and provides an interesting approach to the task.

---

> ### Author Response · Authors · 2018-11-19
> **Response to AnonReviewer1**
>
> Thank you for the thoughtful review and constructive feedback.
>
> Our paper proposes a joint model for localization and repair using pointers, which is novel and the main technical contribution of the paper. Even though it is applied specifically to the variable misuse problem, the idea of using pointers is fundamental and portable to other program repair problems. In particular, all program repair techniques require the bug localization step and pointers seem like an ideal mechanism for this as they can pinpoint a buggy location precisely at the token-level. Other previous works in the program repair literature either use enumerative search for localization, or perform localization at the granularity of lines or depend on external tools for localization (such as compiler error messages for syntactic error localization). In contrast, our proposal to use pointers enables an end-to-end learning based solution. We use the pointer mechanism on top of sequence based encoding of programs, but pointers can be combined naturally with other representations of programs; e.g., trees or graphs.
>
> As we demonstrate in the paper, the previous enumerative approaches assume independence among different predictions that is problematic and leads to poor results. The end-to-end joint localization and repair is an essential step to overcome this issue, and we believe this idea of joint prediction is going to generalize to many other program repair tasks and even program completion tasks.

---

### Official Review · AnonReviewer4 · 2018-11-15
**Interesting Model and Incremental Improvement on Synthetic Datasets and Problematic Problem Definition**

**Rating:** 6
**Confidence:** 5

**Review:**

This paper presents an LSTM-based model for bug detection and repair of a particular type of bug called VarMisuse, which occurs at a point in a program where the wrong identifier is used. This problem is introduced in the Allamanis et al. paper. The authors of the paper under review demonstrate significant improvements compared to the Allamanis et al. approach on several datasets.

I have concerns with respect to the evaluation, the relation of the paper compared to the state-of-the-art in automatic program repair (APR), and the problem definition with respect to live-variable analysis.

My largest concern about both this paper and the Allamanis et al. paper is how it compares to the state-of-the-art in APR in general. There is a large and growing amount of work in APR as shown in the following papers:
[1] L. Gazzola, D. Micucci, and L. Mariani, “Automatic Software Repair: A Survey,” IEEE Transactions on Software Engineering, pp. 1–1, 2017.
[2] M. Monperrus, “Automatic Software Repair: A Bibliography,” ACM Comput. Surv., vol. 51, no. 1, pp. 17:1–17:24, Jan. 2018.
[3] M. Motwani, S. Sankaranarayanan, R. Just, and Y. Brun, “Do automated program repair techniques repair hard and important bugs?,” Empir Software Eng, pp. 1–47, Nov. 2017.

Although the proposed LSTM-based approach for VarMisuse is interesting, it seems to be quite a small delta compared to the larger APR research space. Furthermore, the above papers on APR are not referenced.

The paper under review mostly uses synthetic bugs. However, they do have a dataset from an anonymous industrial setting that they claim is realistic. In such a setting, I would simply have to trust the blinded reviewers. However, the one industrial software project tells me little about the proposed approach’s effectiveness when applied to a significant number of widely-used software programs like the ones residing in state-of-the-art benchmarks for APR, of which there are at least the following two datasets:
[4] C. L. Goues et al., “The ManyBugs and IntroClass Benchmarks for Automated Repair of C Programs,” IEEE Transactions on Software Engineering, vol. 41, no. 12, pp. 1236–1256, Dec. 2015.
[5] R. Just, D. Jalali, and M. D. Ernst, “Defects4J: A Database of Existing Faults to Enable Controlled Testing Studies for Java Programs,” in Proceedings of the 2014 International Symposium on Software Testing and Analysis, New York, NY, USA, 2014, pp. 437–440.

The above datasets are not used or referenced by the paper under review.

My final concern about the paper is the formulation of live variables. A variable is live at certain program points (e.g., program statements, lines, or tokens as called in this paper). For example, from Figure 1 in the paper under review, at line 5 in (a) and (b), object_name and subject_name are live, not just sources.  In the problem definition, the authors say that "V_def^f \subseteq V denotes the set of all live variables", which does not account for the fact that different variables are alive (or dead) at different points of a program. The authors then say that, for the example in Figure 1, "V_def^f contains all locations in the program where the tokens in V appear (i.e., tokens in the Blue boxes), as well as token sources from line 1”. The explanation of the problem definition when applied to the example does not account for the fact that different variables are alive at different program points. I’m not sure to what extent this error negatively affects the implementation of the proposed model. However, the error could be potentially quite problematic.

---

> ### Author Response · Authors · 2018-11-19
> **Response to AnonReviewer4**
>
> Thank you for the thoughtful review and constructive feedback.
>
> We will include a discussion about the differences between our work and the automated program repair (APR) techniques in the literature, as outlined below. The traditional APR approaches differ from our work in the following ways: 1) They require a form of specification of correctness to repair a buggy program, usually as a logical formula/assertion, a set of tests or a reference implementation. 2) They depend on hand-designed search techniques for localization and repair. 3) The techniques are applied to programs which violate the specifications (e.g., a program which fails some tests), that is, to programs which are already known to contain bugs. In contrast, a recent line of research in APR is based on end-to-end learning, of which ours is an instance. Our solution (like some other learning based repair solutions) has the following contrasting features: 1) Our solution does not require any specification of correctness. Instead it learns to fix a common class of errors directly from source code examples. 2) Our solution does not perform enumerative search for localization or repair. We train a neural network to perform localization and repair directly. 3) Our solution is capable of first classifying whether a program has the specific type of bug or not, and subsequently localizing and repairing it.
>
> ManyBugs, IntroClass, and Defects4J are benchmarks designed for test-based program repair techniques. The bugs relate to the expected specification of individual programs (captured through test cases of the program) and the nature of bugs vary from program to program. These benchmarks are therefore suitable to evaluate repair techniques guided by test executions. Learning based solutions like ours focus on common error types so that it is possible for a model to generalize across programs, and work directly on embeddings of source code.
>
> Thank you for your comment about the variable liveness. We misused the term of live variables, and we will update the paper accordingly. The V_def^f set contains all variables defined in a function f, including the function arguments; in this way constructing a set of all variables that can be used within the scope. We construct one V_def^f set per function, representing a set of candidate variables for fixing bugs in that function. In this way, the V_def^f set is a (safe) over-approximation of the in-scope variables at each program location. The over-approximation can lead to predicting an undefined variable as a repair, however, this is not an error and model over time learns not to predict undefined variables. The V_def^f set is not constrained to only live variables; as there are cases when solution to a bug is using a variable that is defined in the scope but not live (not used elsewhere), e.g., subject_name variable in Figure 1a. Regarding your comment about Figure 1, in the blue boxes we show variable usages, not V_def^f set.
>
> We want to clarify that the examples in our industrial dataset (Section 4.4) are not from a single industrial project. The examples do come from multiple software projects.
>
> Please let us know if this helped clarify the confusion regarding the problem definition of candidate variable set and the relationship with previous APR work and the more recent neural program repair approaches (ours, Allamanis et. al and others).

---

> > ### Comment · AnonReviewer4 · 2018-11-22
> > **Related Work, Definition Correction, Industrial Dataset, Implementation Availability**
> >
> > I appreciate the authors adding the related work I noted as missing in the paper and distinguishing their work more from the state of the art.
> >
> > The explanation of V_def^f in this response is more helpful than the one line explanation given in the revised paper. In particular, I find the following sentences useful: "The V_def^f set contains all variables defined in a function f, including the function arguments; in this way constructing a set of all variables that can be used within the scope. We construct one V_def^f set per function, representing a set of candidate variables for fixing bugs in that function." I encourage the authors to clarify their definition of V_def^f further in the paper, as they did in this response.
> >
> > I think it is important for the authors to clarify that their industrial dataset includes multiple projects in the paper, not just in this response.
> >
> > Are the authors going to make an implementation of their approach available for others to build upon (e.g., base new approaches on) or improve upon? I believe such practices of releasing implementations help to enhance the science and accelerates the rate of that improvement.

---

> > > ### Author Response · Authors · 2018-11-27
> > > **Suggestions incorporated**
> > >
> > > We thank the reviewer for useful suggestion to further update the paper with clarification on V_def^f set and industrial dataset; and we have updated the paper accordingly in the new revision.
> > >
> > > We do intend to make our code available to others. We believe that the implementation details as described in the paper are also sufficient for reproducing our technique and results.
> > >
> > > Please let us know in case you have any further concerns, and/or suggestions of how we can further improve the paper.

---

### Author Response · Authors · 2018-11-19
**Feedback Incorporated In New Paper Version**

Thanks to all the reviewers for their helpful and constructive feedback. We have uploaded a new paper revision to address the comments and feedback:

1. Added a new section 4.4 on evaluation of the model in practice on realistic bugs.
2. Added a discussion about key differences with the previous work: DeepBugs and Sk_p (Section 2).
3. Added a discussion on performance comparison with enumerative approaches (Section 4.1).
4. Changed the problem definition to clarify the definition of V_def set (Section 3.1).
5. Added a discussion on the relationship between previous automated program repair (APR) work based on tests/specifications and neural program repair approaches (Section 2).

---

### Meta-Review · Area_Chair1 · 2018-12-17
**Lacking novelty, but strong results and evaluation, well-written paper**

**Confidence:** 3
**Recommendation:** Accept (Poster)

**Metareview:**

This paper provides an approach to jointly localize and repair VarMisuse bugs, where a wrong variable from the context has been used. The proposed work provides an end-to-end training pipeline for jointly localizing and repairing, as opposed to independent predictions in existing work. The reviewers felt that the manuscript was very well-written and clear, with fairly strong results on a number of datasets.

The reviewers and AC note the following potential weaknesses: (1) reviewer 4 brings up related approaches from automated program repair (APR), that are much more general than the VarMisuse bugs, and the paper lacks citation and comparison to them, (2) the baselines that were compared against are fairly weak, and some recent approaches like DeepBugs and Sk_p are ignored, (3) the approach is trained and evaluated only on synthetic bugs, which look very different from the realistic ones, and (4) the contributions were found to be restricted in novelty, just uses a pointer-based LSTM for locating and fixing bugs.

The authors provided detailed comments and a revision to address and clarify these concerns. They added an evaluation on realistic bugs, along with differences from DeepBugs and Sk_p, and differences between neural and automated program repair. They also added more detail comparisons, including separating the localization vs repair aspects by comparing against enumeration. During the discussion, the reviewers disagree on the "weakness" of the baseline, as reviewers 1 and 4 feel it is a reasonable baseline as it builds upon the Allamanis paper. They found, to different degrees, that the results on realistic bugs are much more convincing than the synthetic bug evaluation. Finally, all reviewers agree that the novelty of this work is limited.

Although the reviewers disagree on the strength of the baselines (a recent paper) and the evaluation benchmarks, they agreed that the results are quite strong. The paper, however, addressed many of the concerns in the response/revision, and thus, the reviewers agree that it meets the bar for acceptance.